# Early Pregnancy Modulates Expression of the Nod-like Receptor Family in Lymph Nodes of Ewes

**DOI:** 10.3390/ani12233285

**Published:** 2022-11-25

**Authors:** Zhenyang Zhao, Yuanjing Li, Jianhua Cao, Hongxu Fang, Leying Zhang, Ling Yang

**Affiliations:** School of Life Sciences and Food Engineering, Hebei University of Engineering, Handan 056038, China

**Keywords:** lymph node, NOD-like receptor, pregnancy, sheep

## Abstract

**Simple Summary:**

NOD receptors mediate adaptive immune responses and immune tolerance. In this study, the results demonstrated that early pregnancy increased expression of NOD1, CIITA, NLRP1, NLRP3 and NLRP7. However, early pregnancy inhibited NAIP expression in maternal lymph nodes. There is immune regulation of the lymph nodes, which is necessary during ovine gestation.

**Abstract:**

NOD receptors (NLRs) mediate adaptive immune responses and immune tolerance. Nevertheless, it is not clear if gestation modulates the NLR signaling pathway in lymph nodes of ewes. In this study, lymph nodes of ewes were collected at day 16 of the estrous cycle, and at days 13, 16 and 25 of gestation (*n* = 6 for each group). RT-qPCR, Western blot and immunohistochemistry analysis were used to analyze the expression of the NLR family, including NOD1, NOD2, CIITA, NAIP, NLRP1, NLRP3 and NLRP7. The data showed that early gestation enhanced expression of NOD1, CIITA, NLRP1, NLRP3 and NLRP7 mRNA, as well as proteins at day 16 of gestation, and the expression levels of NOD2, CIITA, NLRP1 and NLRP7 were higher at days 13 and 25 of gestation than day 16 of the estrous cycle. However, NOD1 expression was lower on days 13 and 25 of gestation compared to day 16 of the estrous cycle, and early gestation suppressed NAIP expression. In summary, early pregnancy modulated expression of the NLR family in ovine lymph nodes, which participates in immune regulation, and this modulation may be necessary for pregnancy establishment in ewes.

## 1. Introduction

Nucleotide-binding oligomerization domain (NOD) receptors (NLRs) are involved in immune responses through specifically triggering numerous signaling pathways, including the nuclear factor kappa B (NF-κB) pathway [1]. NOD1 and NOD2 belong to the NLR family of innate immune proteins that are related to priming adaptive immune responses and defining immune tolerance [2]. Major histocompatibility complex (MHC) class II transactivator (CIITA) is involved in MHC class II gene transactivation, which plays key roles in activating the adaptive immune system [3]. NLR family, pyrin domain-containing (NLRP) proteins are NLR innate immune sensors, and NLRP3 initiates the assembly process of a multiprotein complex to regulate multiple host defense pathways [4]. Neuronal apoptosis inhibitor protein (NAIP) is abundantly expressed in anti-inflammatory macrophages, which is involved in macrophage differentiation and modulating innate and adaptive immunity [5]. NLRP3 is increased in the myometrium with labor, which is related to regulating inflammation-induced pro-labor mediators [6]. It is reported that NLRs are involved in endometrial remodeling and placental development during early pregnancy [7].

During pregnancy, various immune effectors participate in the establishment of maternal tolerance toward the semi-allogeneic fetus, which set up a favorable humoral immunity [8]. In addition, some modulators of the maternal–fetal interface and humoral immune components play key roles in maintaining normal pregnancy in sheep [9]. Interferon-tau (IFNT) exerts paracrine and endocrine actions to inhibit luteolytic pulses of prostaglandin (PG) F2α and maintain corpus luteum function in ruminants, which modulate the innate immune events to avoid allogenic conceptus rejection from females during early gestation [10]. IFNT is involved in hormonal communication with the maternal immune system, which is helpful for the fetus in ruminants [11]. It has been reported that IFNT and progesterone regulate expression of interferon-stimulated genes (ISGs), progesterone receptor (PGR) isoforms and progesterone-induced blocking factor (PIBF) variants in the ovine immune organs, including the bone marrow [12,13], the thymus [14,15], the spleen [16,17,18] and lymph nodes [19,20,21], during early pregnancy.

Regulation of innate cell microenvironments within lymph nodes plays critical roles in generating adaptive responses [22]. There is a modulation of the maternal immune system during successful pregnancy, and the immune functions of lymph nodes are different between them around the reproductive tract and others in ewes [23]. Progesterone and its metabolites (PGRs and PIBF1) in lymphatic endothelial cells modulate tumor necrosis factor (TNF) α and interferon-γ (IFN-γ) production, which are critical in regulating immune tolerance during gestation [24]. PG synthases, including cyclooxygenase 1 (COX-1), COX-2, a PGE synthase (PTGES), and a PGF synthase (Aldo-keto reductase family 1, member B1, AKR1B1), as well as interleukin-5 (IL-5) and IL-10 are increased in lymph nodes, but TNFβ and IL-2 downregulated during early pregnancy in sheep [21,25]. Proteins of melatonin receptor 1, gonadotropin-releasing hormone and its receptor, prolactin and its receptor, and a cluster of differentiation 4 are increased in the lymph nodes during ovine early gestation [26,27,28]. Furthermore, gestation regulates expression of the Toll-like receptors (TLRs), TNF receptor-associated factor 6, IL-1 receptor-associated kinase 1 and myeloid differentiation primary response gene 88 [29], as well as the complement components C1q, C1r, C1s, C2, C3, C4a, C5b and C9 and the NF-κB family in maternal lymph nodes [30,31].

NAIP is expressed in lymph nodes, which is implicated in immune response [32]. It is supposed that, in maternal lymph nodes, expression of NLR family may be modulated by early pregnancy. The objective of this study is to analyze the expression of NOD1, NOD2, CIITA, NAIP, NLRP1, NLRP3 and NLRP7 during early gestation in ewes. The data will be helpful for comprehending the modulation of immune tolerance during early gestation and thus help to improve the reproduction rate in ruminants.

## 2. Materials and Methods

### 2.1. Animals and Experimental Design

The experiment was carried out during the normal breeding season. A total of 24 ewes (1.5 to 2.0 years old; Small-tail Han sheep) were randomly and equally divided into four groups (based on weight and body condition score), and joined with either three adult intact rams (pregnant group) or a vasectomized ram (nonpregnant group). Ewes were inspected twice daily for mating marks, and ewes that showed a new mating mark (considered as day 0). Inguinal lymph nodes were collected on days 13, 16 and 25 for pregnant animals, and lymph node collection in nonpregnant group was assigned on day 16 after showing a new mating mark, as described previously [19]. Days 13, 16, and 25 of pregnancy and day 16 the estrous cycle were chosen (*n* = 6 for each group), because the maternal lymph nodes were under the different effects of IFNT and/or progesterone, or not on these days [26]. Pregnancy was confirmed by the presence of a normal conceptus in the uterine lumen. Longitudinal cross sections of the lymph nodes were immersion fixed in fresh 4% buffered paraformaldehyde for subsequent immunohistochemical analysis. In addition, transverse pieces of the lymph nodes were frozen for subsequent mRNA isolation and protein analyses.

### 2.2. RNA Extraction and RT-qPCR Assay

Total RNA was extracted and isolated from the frozen samples of the lymph nodes using TRNzol Universal Reagent (DP424; Tiangen Biotech Co., Ltd., Beijing, China), as per the manufacturer’s instructions. A 1-μg aliquot of total RNA was DNase-I treated and reverse transcribed into first-strand cDNA using a FastQuant RT kit with gDNase (KR106; Tiangen Biotech). The specified primers were designed based on the sequences in the NCBI database (http://www.ncbi.nlm.nih.gov/ (accessed on 2 May 2021)) for the ovine genes of NLR family, and synthesized by Shanghai Sangon Biotech Co., Ltd. (Shanghai, China). Primer details are provided in Table 1, and the specificity of the primers were verified using non-template controls. There was an efficiency of 96% ± 3% for the mean primer. The annealing temperatures were 60.5 °C for NOD1 and CIITA, or 62 °C for NOD2, or 59.5 °C for NAIP, or 60 °C for NALP1, or 59 °C for NLRP3, or 61 °C for NLRP7. Glyceraldehyde phosphate dehydrogenase gene (GAPDH) was analyzed in parallel in all target genes. The 2^−ΔΔCt^ analysis method was used to calculate the relative values for the target genes [33], and the relative levels of the mRNA transcripts were normalized using the mean cycle threshold values from the group of day 16 of the estrous cycle.

### 2.3. Western Blot Analysis

Lymph node tissues were homogenized using RIPA lysis buffer supplemented with phosphatase and protease inhibitor on ice for 15 min, and a bicinchoninic acid assay kit (Tiangen Biotech) was used to determine the protein concentrations. Equal amounts of protein (10 μg per well) were separated by SDS-PAGE using 12% polyacrylamide gel and transferred electrophoretically to methanol-activated polyvinyl difluoride membranes (Millipore, Bedford, MA, USA) for immunoblotting. Nonfat milk (5%, *w*/*v*) was used for blocking nonspecific protein-binding at 4 °C overnight. The membranes were incubated with primary antibodies in a 1:1000 dilution at 4 °C overnight. The primary antibodies included a mouse anti-NOD1 monoclonal antibody (Santa Cruz Biotechnology, Santa Cruz, CA, USA, sc-398696), a mouse anti-NOD2 monoclonal antibody (Santa Cruz Biotechnology, sc-56168), a mouse anti-CIITA monoclonal antibody (Santa Cruz Biotechnology, sc-13556), a rabbit anti-NAIP polyclonal antibody (Abcam, Cambridge, UK, ab25968), a mouse anti-NLRP1 monoclonal antibody (Santa Cruz Biotechnology, sc-390133), a mouse anti-NLRP3 monoclonal antibody (Santa Cruz Biotechnology, sc-134306), and a mouse anti-NLRP7 monoclonal antibody (Santa Cruz Biotechnology, sc-377190). The target proteins were analyzed by an ECL Western blot detection kit (Tiangen Biotech), and visualized with X-ray films (Fujifilm, Tokio, Japan). Quantity One V452 (Bio-Rad Laboratories, Hercules, CA, USA) was used to analyze THE densitometry values of the blots. The relative intensity of the blots was calculated and normalized with a reference protein (GAPDH) using a GAPDH antibody (Santa Cruz Biotechnology, sc-47724).

### 2.4. Immunohistochemistry Analysis

Immunohistochemistry was used for detection of the NOD2 and NLRP7 proteins, and NOD2 was the representative for NODs, and NLRP7 was the representative for the NLRPs. The procedures were described previously by Zhang et al. [19] using primary antibodies specific to NOD2 (sc-56168), or to NLRP7 (sc-377190). Slides were imaged using a light microscope (Nikon Eclipse E800, Tokyo, Japan) equipped with a DP12 digital camera. The images were examined independently by 4 experienced observers with the similar observations for all cases. The immunostaining intensities of the lymph node samples from the different ewes were checked through the images in a blinded manner. Staining intensities for NOD2 or NLRP7 protein were calculated by assigning an immunoreactive intensity score using a scale of 0 to 3.

### 2.5. Statistical Analysis

The experimental design was completely randomized with ewe as the experimental unit. Statistical analysis was performed using the mixed procedure of SAS (version 9.4; SAS Institute Inc., Cary, NC, USA). Day and status (cyclic or pregnant), and day × status interactions were included in our model. Each group consisted of three replicates. All data were expressed as the means ± standard deviation (SD), and all tests were deemed significant if *p* < 0.05.

## 3. Results

### 3.1. Expression of NOD1, NOD2, CIITA, NAIP, NLRP1, NLRP3 and NLRP7 mRNA in Lymph Nodes

It is shown in Figure 1 that the expression levels of NOD1, CIITA, NLRP1, NLRP3 and NLRP7 mRNA were the highest at day 16 of gestation (*p* < 0.05), and expression levels of CIITA, NLRP1 and NLRP7 were higher at day 25 of gestation compared to day 16 of the estrous cycle (*p* < 0.05). However, the NOD1 level was lower at days 13 and 25 of gestation compared to day 16 of the estrous cycle (*p* < 0.05), and there was no significant difference in the NLRP3 level among the lymph nodes from day 16 of nonpregnancy and from days 13 and 25 of gestation. The NOD2 mRNA expression value was higher at days 16 and 25 of gestation compared to day 13 of gestation and day 16 of nonpregnancy (*p* < 0.05), and the NOD2 mRNA level was lower at day 16 of nonpregnancy compared to other groups. Furthermore, pregnancy suppressed the NAIP mRNA expression compared to day 16 of nonpregnancy (*p* < 0.05).

### 3.2. Expression of NOD1, NOD2, CIITA, NAIP, NLRP1, NLRP3 and NLRP7 Proteins in Lymph Nodes

Expression levels of NOD1, CIITA, NLRP1, NLRP3 and NLRP7 proteins peaked at day 16 of gestation (Figure 2; *p* < 0.05), and the CIITA, NLRP1 and NLRP7 proteins were enhanced on day 25 of gestation compared to that on day 16 of nonpregnancy (*p* < 0.05). It was undetected for NLRP1 protein on day 16 of nonpregnancy and day 13 of gestation, and NLRP7 protein at day 16 of nonpregnancy. Nevertheless, the NOD1 protein value was higher on day 16 of nonpregnancy than days 13 and 25 of gestation (Figure 2; *p* < 0.05), and expression values of NLRP3 protein was not significantly different among day 16 of nonpregnancy and days 13 and 25 of gestation. Furthermore, early gestation induced expression of NOD2 protein, and the NOD2 protein level was higher on days 16 and 25 of gestation compared to day 13 of gestation and day 16 of nonpregnancy (*p* < 0.05). On the other hand, early pregnancy inhibited expression of NAIP protein (Figure 2; *p* < 0.05).

### 3.3. Immunohistochemistry for NOD2 and NLRP7 Proteins in Maternal Lymph Nodes

NOD2 and NLRP7 proteins were located in the subcapsular sinus and lymph sinuses (Figure 3). For the negative control, the lymph nodes from day 16 of nonpregnancy, and lymph nodes from days 13, 16 and 25 of gestation, the staining intensities for NOD2 protein were 0, 1, 2, 3 and 3, while the staining intensities for NLRP7 protein were 0, 0, 1, 2 and 1, respectively (Figure 3). The staining intensity was as follows: 0 = negative; 1 = weak; 2 = strong; 3 = stronger.

## 4. Discussion

In this study, expression of NOD1 in maternal lymph nodes peaked at day 16 of gestation but was downregulated at day 25 of gestation. NOD1 is profusely expressed by immune and non-immune cells, and associated with many metabolic pathways that are related to activation of immune responses [34]. The NOD1 expression level in the placental villi from recurrent spontaneous abortion females is higher compared to normal pregnancy, and NOD1 inhibited the invasion of trophoblast cells during gestation [35]. In addition, human first-trimester trophoblasts express NOD1, and NOD1 activation induces a proinflammatory cytokine profile, and triggers preterm delivery [36]. Furthermore, administration of FK565 induces a higher expression level of NOD1 in fetal vascular tissues and at the maternal–fetal interface, which result in intrauterine fetal growth restriction (IUGR) and death in pregnant mice [37]. Moreover, a higher level of NOD1 protein is related to development of IUGR, which leads to fetal and neonatal morbidity [38]. On the other hand, there is a higher level of NOD1 in trophoblasts in normal pregnancy, which has direct effects on maternal–fetal communication [39]. Therefore, changes in expression of NOD1 may be associated with maternal–fetal communication, which may be beneficial for pregnancy establishment in sheep.

Our data revealed that NOD2 was upregulated in the maternal lymph node during early pregnancy, and the NOD2 protein was located in the subcapsular sinus and lymph sinuses. NOD2 can recognize danger signals in the cells and plays an essential role in maintaining the innate immune response [40]. First trimester trophoblast cells express NOD2 that contribute to recognizing and responding to invasive intracellular pathogens [41]. In addition, the NOD2 level in decidual stromal cell from females with unexplained recurrent spontaneous abortion is lower compared to normal pregnancy, suggesting that the NOD2 protein is required for sustaining normal pregnancy in humans [42]. Furthermore, common TLR4 and NOD2 gene variants alter the maternal inflammatory responses, which damage the inflammatory response to endotoxin, and are associated with severe hypertensive disorders in pregnancy [43]. Expression of TLR4 in maternal lymph node is also increased during early gestation, which is related to modulating the maternal innate immune response [29]. These reports are almost consistent with our findings in maternal lymph nodes. Therefore, the upregulation of NOD2 may be beneficial for modulation of innate immune responses.

In this study, the expression level of CIITA increased during early gestation. CIITA interacts with itself and other proteins to regulate multiple immune responses [44]. There is an increased expression of CIITA protein in endometria during early gestation, and the CIITA protein is localized in the basal layer of the endometrial luminal epithelial cells in pigs [45]. In addition, IFN-γ classically induces expression of CIITA and regulates the MHC II molecule in many cell types to invoke protective immunity and maintain tolerance to beneficial antigens [46]. Furthermore, expression of IFN-γ peaked on day 16 of gestation, which is related to immunoregulation of maternal lymph nodes in sheep [25]. Moreover, CIITA was expressed in trophectoderm at the inner cell mass side of the embryo, but knockdown of CIITA expression in in vitro-derived embryos disrupts the development of an embryo into the blastocyst stage, indicating that CIITA is required for preimplantation development bovine embryos [47]. These findings indicate that CIITA is essential for pregnancy. Therefore, the upregulation of CIITA during pregnancy may be related to maternal immune modulation at the implantation period in sheep.

It was found in this study that expression of NAIP was decreased during early gestation. The NAIP/NLR-family CARD-containing protein 4 inflammasome forms through oligomerization after ligand recognition by NAIP sensor proteins, which results in a strong autoinflammatory response that has adverse consequences for the host [48]. NAIP mRNA is expressed in human placentas, and there is an increase in its expression level in the placental tissue samples from term placentas compared to from the first trimester [49]. In addition, NAIP has critical effects on modulating the phenotype of the trophoblast cell, which is implicated in the progression of preeclampsia development [50]. Furthermore, NAIP protein in reactive lymphoid hyperplasia lymph nodes is higher than that in normal lymph node samples [32]. Therefore, the downregulation of NAIP may be involved in attenuating the maternal autoinflammatory response, and beneficial for pregnancy establishment and maintenance in ewes.

Our data showed that there was a peak in expression level of NLRP1 at day 16 of pregnancy, but which then was downregulated. NLRP1 is expressed in immune cells, and also is an immune sensor that can detect diverse pathogen- and non-pathogen-encoded activities to trigger an innate immune response [51,52]. There is an increase in NLRP1 in the uterus of abortion rats, and a high expression level of NLRP1 is related to placental dysfunction and adverse pregnancy [53]. In addition, expression of NLRP1 is significantly higher in the placenta from preeclamptic pregnant females compared to normal pregnant females, but vitamin D treatment can decrease NLRP1 expression in the placental explants from women with preeclampsia [54]. Furthermore, there are separate expression profiles of NALP1 in human tissues, and lymph nodes and tracheas have the highest expression levels of NALP1 than other tissues, suggesting that NALP1 plays a site-specific role [51,55]. Therefore, the peak of NALP1 may be involved in immune modulation of maternal lymph nodes, but downregulation at day 25 of pregnancy may be beneficial for weakening immune responses.

Our results revealed that the NLRP3 expression level peaked at day 16 of gestation, but which then was downregulated. NLRP3 is implicated in many physiological and cellular processes, including metabolism and immune responses [56]. Placenta from women with preeclampsia shows an upregulation of NLRP3 compared to normotensive pregnant women, and higher expression of NLRP3 inflammasome exaggerates the inflammatory state in preeclampsia [57]. In addition, damage-associated molecular patterns induce production of NLRP3, which plays a key role in sterile inflammation and is implicated in pregnancy dysfunction, including preeclampsia [58]. However, NLRP3 improves embryo implantation during the implantation window [59]. Furthermore, decidualization improves the production of immunomodulatory factors, including the NLRP3 inflammasome, that are in favor of decidualization and placentation [60]. Therefore, the upregulation of NLRP3 may be related to the initiation of embryo implantation, but downregulation of NLRP3 may be beneficial for pregnancy maintenance in ewes.

It was showed in this study that early pregnancy enhanced NLRP7 expression, and the NLRP7 protein was located in the subcapsular sinus and lymph sinuses. NLRP7 is abundant in endometrial tissues and decidual macrophages, which participate in decidualization and macrophage differentiation, and play key roles in successful pregnancy [61]. There is an upregulation of NLRP7 in first-trimester endometrium compared to non-pregnant women, and knockdown or overexpression of NLRP7 decreases or increases decidualization [62]. In addition, the NLRP7 protein is mainly located in pre-implantation embryos, and NLRP7 knockdown has adverse effects on the parthenogenetic and in vitro fertilization embryo development in sheep [63]. Furthermore, NLRP7 has effects on trophoblast lineage differentiation, which are related to early embryonic development [64]. These reports support the idea that the upregulation of NLRP7 is favorable for pregnancy. Therefore, the upregulation of NLRP7 may be related to regulation of the maternal immune response, and NLRP7 may be used for enhancing the pregnancy rate of ewes.

## 5. Conclusions

Early pregnancy enhanced expression of NOD2, CIITA, NLRP1 and NLRP7 in maternal lymph nodes, and expression levels of NOD1, NOD2, CIITA, NLRP1, NLRP3 and NLRP7 peaked on day 16 of gestation. However, NOD1 was downregulated at days 13 and 25 of gestation, and NAIP decreased during early gestation. In addition, the NOD2 and NLRP7 proteins were limited to the subcapsular sinus and lymph sinuses. Therefore, early pregnancy has effects on expression of the NLR family in ovine lymph node, which may be partly involved in the gestation recognition signal (IFNT) and the maternal immune regulation, and modulation of the NLR family may be used for improving the reproduction rate in ruminants.

## Figures and Tables

**Figure 1 animals-12-03285-f001:**
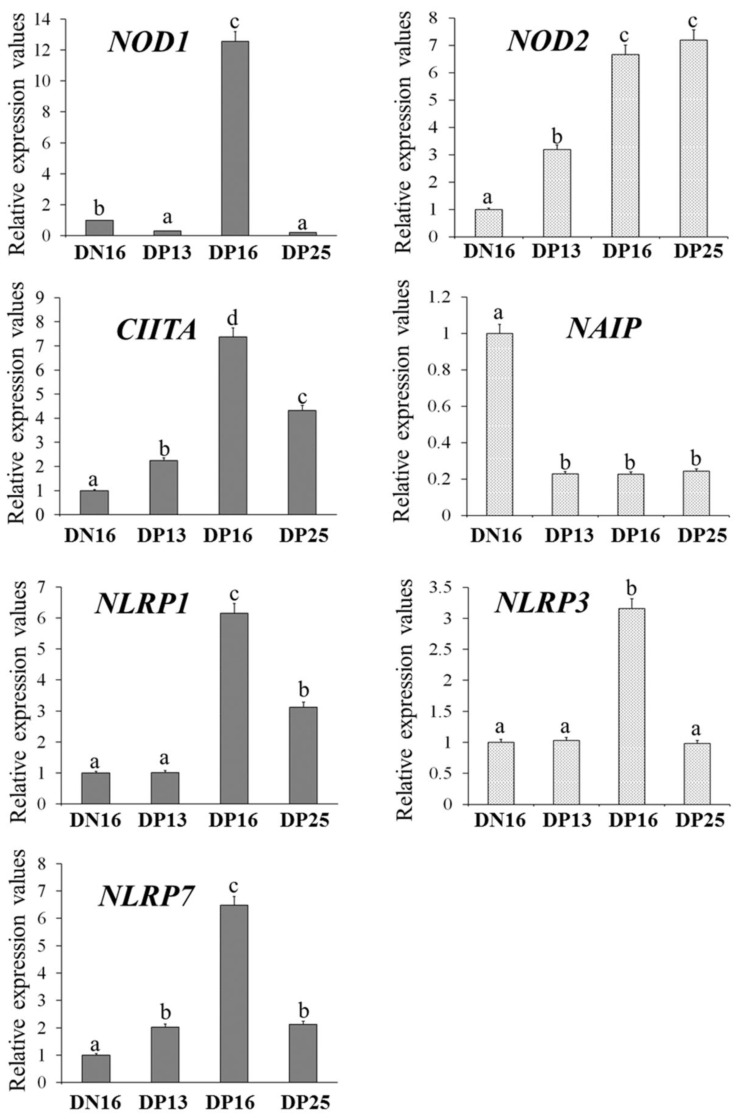
Relative expression values of NOD1, NOD2, CIITA, NAIP, NLRP1, NLRP3 and NLRP7 mRNA in the lymph nodes. Note: DN16 = day 16 of nonpregnancy; DP13 = day 13 of gestation; DP16 = day 16 of gestation; DP25 = day 25 of gestation. Significant differences (*p* < 0.05) are indicated by different letters.

**Figure 2 animals-12-03285-f002:**
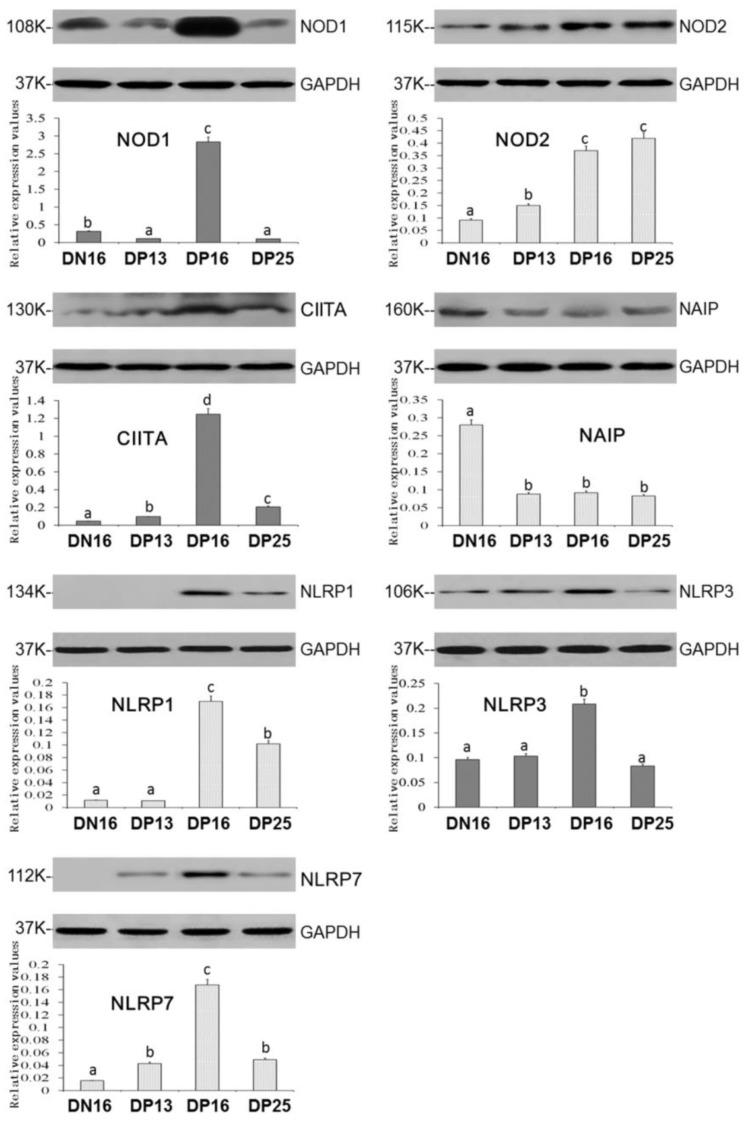
Expression of NOD1, NOD2, CIITA, NAIP, NLRP1, NLRP3 and NLRP7 proteins in the lymph nodes. Note: DN16 = day 16 of nonpregnancy; DP13 = day 13 of gestation; DP16 = day 16 of gestation; DP25 = day 25 of gestation. Significant differences (*p* < 0.05) are indicated by different letters within the same color column (original Western blot figures are in Appendix A).

**Figure 3 animals-12-03285-f003:**
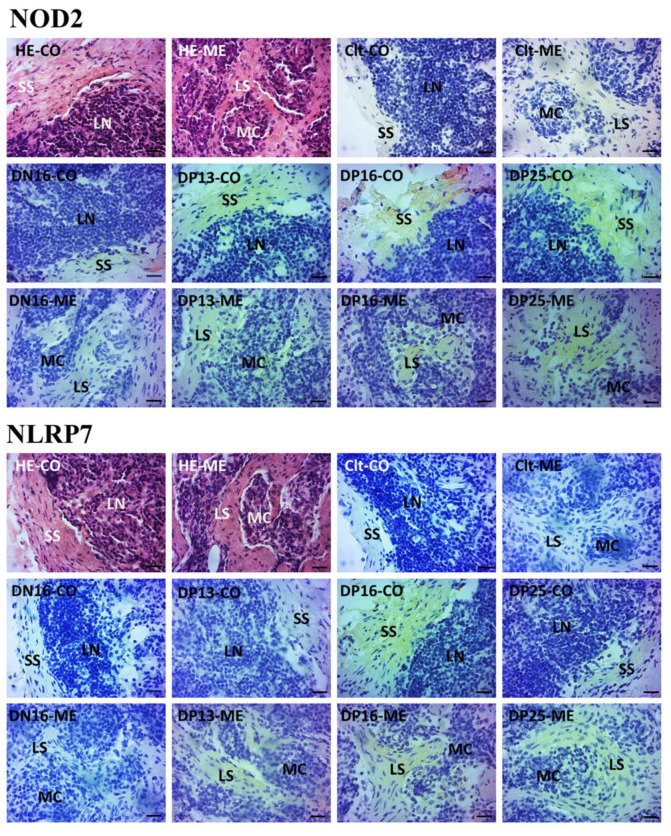
Representative immunohistochemical localization of NOD2 and NLRP7 proteins in the lymph nodes. The lymph node is divided into the outer cortex (CO) and the inner medulla (ME). Lymph enters the convex through the subcapsular sinus (SS) around the lymphoid nodules (LN) and flows into the medulla through the lymph sinus (LS) around the medullary cord (MC). Note: HE = stained by hematoxylin and eosin; Clt = negative control; DN16 = day 16 of nonpregnancy; DP13 = day 13 of gestation; DP16 = day 16 of gestation; DP25 = day 25 of gestation. Bar = 20 μm.

**Table 1 animals-12-03285-t001:** The primers used.

Gene	Primer	Sequence	Size (bp)	Accession Numbers
NOD1	Forward	CCTTGGCTGTCAGAGATTGGCTTC	94	XM_042248630.1
	Reverse	GCTTCTGGCTGTATCTGCTCACTG		
NOD2	Forward	TGCCATCCTCGCTCAGACATCTC	117	XM_042231601.1
	Reverse	CAGCCACACTGCCCTCTTTGC		
CIITA	Forward	GCACCTCCTTCCAGTTCCTTGTTG	119	XM_042239890.1
	Reverse	CCTGTCCCAGTCCCTGAGATCG		
NAIP	Forward	TTGTCCAGCAGTGTCAGCATCTTC	82	XM_012096791.3
	Reverse	ATTTCCACCACGCTGTCATCATCC		
NLRP1	Forward	AAGGAGGTGACCGAGATGCTGAG	143	XM_012185551.4
	Reverse	TGCCGCTTGAGTGAGGATGTATTG		
NLRP3	Forward	CTCTGGTTGGTCAGTTGCTGTCTC	81	XM_042250402.1
	Reverse	GGTCAGGGAATGGTTGGTGCTTAG		
NLRP7	Forward	GCCTGCTACTCGTTCATCCATCTC	90	XM_004015893.5
	Reverse	CCCTTCCTCCTCCTGCTCTTCC		
GAPDH	Forward	GGGTCATCATCTCTGCACCT	176	NM_001190390.1
	Reverse	GGTCATAAGTCCCTCCACGA		

## Data Availability

Not applicable.

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
