# Peer review of "Early Pregnancy Modulates Expression of the Nod-like Receptor Family in Lymph Nodes of Ewes"

_animals, 2022, doi:10.3390/ani12233285_

Round 1
Reviewer 1 Report
The present manuscript analyses the expression of Nucleotide-binding oligomerization domain receptors (Nod-like receptor family; NLRs) in lymph nodes of early pregnant ewes in order to know whether these expression profiles are affected by different stages of pregnancy. For this purpose, 24 ewes were used being randomly divided in four groups as follows: 13, 16 and 25 pregnant animals and a control group consisting non pregnant ewes at day 16 of the cycle. The mRNA and protein expression of NOD1, NOD2, CIITA, NAIP, NLRP1, NLRP3 and NLRP7 in lymph nodes was analyzed by RT-PCR and western blot, respectively. In addition, detection of NOD2 and NLRP7 in lymph nodes by immunohistochemistry was performed. The obtained results seem to indicate that expression of the different NLRs evaluated changed according to the pregnancy stage favouring immunotolerant environment at uterine level and therefore contributing to an adequate pregnancy establishment and progression. The paper is interesting since all efforts made for improving the knowledge about physiology and immunology of early pregnancy stages are important for the optimization of ovine reproductive management. However, and despite the very interesting topic of the manuscript there are several aspects, that should be modified by the authors before the manuscript is ready for its publication. The manuscript needs to be deeply revised and improved before being ready for publication. Recommendation: Major revision
Comments to the authors:
- General comments: The general impression is that the authors have written the manuscript very quickly and have not devoted the necessary effort to prepare an article worthy of publication. From the title to the conclusions, the manuscript is poorly cared for and as a result the scientific quality is diminished. I regret that with such an interesting topic and with the results obtained, the authors have not prepared a better manuscript.
- Title: Please, revise the tittle. It seems that and “in” is missing “…family in Lymph nodes of ewes”. Change if convenient.
- Introduction: This section should be modified in order to focus better in the importance analyzed NLRs in the context of early pregnancy in ewes. In the current version of the manuscript just same lines are dedicated to this NLRs while the rest of the discussion is more related with other immune aspects of pregnancy.
In addition, the objective of the manuscript is very poorly described.
- Materials and Methods: This section is also very poor. There is an important lack of relevant information in all points included in this section. Even in the case that reference to previous work is made a brief description of the procedures should be included. The description of the experimental groups is confusing and is not written in a correct way. Regarding immunohistochemistry, the reason for selecting NOD2 and NLRP7 should be included. In addition, description of the handling of lymph nodes for the different analysis techniques performed should be included.
Information about image obtaining and analysis should also be included.
Finally a more detailed description of the statistical analysis should be also provided.
- Results: Please provide some details about how relative expression of mRNA and proteins were obtained.
Please revise the paragraph describing the results about immunohistochemistry. In the current version of the manuscript this information is not adequate.
- Discussion: Although relevant information is included, this section is repetitive. The authors have made a very simplistic approach to this part of the work. Again, despite the importance of the topic and the interesting results obtained, the authors have made little effort to discuss these results in an adequate way to arouse the interest to the scientific community. As in the rest of the manuscript, it seems that the authors have not dedicated to this manuscript the necessary work and effort to make it scientifically relevant.
- Conclusion: Page 7; Lines 250 and follows: I would recommend to revise also the conclusion section. In its present form, it is just a brief description of the obtained results and it does not make clear the relevance that the results obtained could have on a practical level on ovine reproduction.
Author Response
Response to Reviewer 1 Comments
The present manuscript analyses the expression of Nucleotide-binding oligomerization domain receptors (Nod-like receptor family; NLRs) in lymph nodes of early pregnant ewes in order to know whether these expression profiles are affected by different stages of pregnancy. For this purpose, 24 ewes were used being randomly divided in four groups as follows: 13, 16 and 25 pregnant animals and a control group consisting non pregnant ewes at day 16 of the cycle. The mRNA and protein expression of NOD1, NOD2, CIITA, NAIP, NLRP1, NLRP3 and NLRP7 in lymph nodes was analyzed by RT-PCR and western blot, respectively. In addition, detection of NOD2 and NLRP7 in lymph nodes by immunohistochemistry was performed. The obtained results seem to indicate that expression of the different NLRs evaluated changed according to the pregnancy stage favouring immunotolerant environment at uterine level and therefore contributing to an adequate pregnancy establishment and progression. The paper is interesting since all efforts made for improving the knowledge about physiology and immunology of early pregnancy stages are important for the optimization of ovine reproductive management. However, and despite the very interesting topic of the manuscript there are several aspects, that should be modified by the authors before the manuscript is ready for its publication. The manuscript needs to be deeply revised and improved before being ready for publication. Recommendation: Major revision
Response: Thanks.
Comments to the authors:
General comments: The general impression is that the authors have written the manuscript very quickly and have not devoted the necessary effort to prepare an article worthy of publication. From the title to the conclusions, the manuscript is poorly cared for and as a result the scientific quality is diminished. I regret that with such an interesting topic and with the results obtained, the authors have not prepared a better manuscript.
Response: The manuscript had been revised throughout.
Title: Please, revise the tittle. It seems that and “in” is missing “…family in Lymph nodes of ewes”. Change if convenient.
Response: The tittle had been revised. Please see the tittle.
Introduction: This section should be modified in order to focus better in the importance analyzed NLRs in the context of early pregnancy in ewes. In the current version of the manuscript just same lines are dedicated to this NLRs while the rest of the discussion is more related with other immune aspects of pregnancy.
Response: Introduction section had been revised. Please see the Introduction section.
In addition, the objective of the manuscript is very poorly described.
Response: The objective had been revised. Please see the objective in the Introduction section.
Materials and Methods: This section is also very poor. There is an important lack of relevant information in all points included in this section. Even in the case that reference to previous work is made a brief description of the procedures should be included. The description of the experimental groups is confusing and is not written in a correct way. Regarding immunohistochemistry, the reason for selecting NOD2 and NLRP7 should be included. In addition, description of the handling of lymph nodes for the different analysis techniques performed should be included.
Response: Materials and Methods section had been revised. Please see the Materials and Methods section.
Information about image obtaining and analysis should also be included.
Response: Information about image obtaining and analysis had been added.
Finally a more detailed description of the statistical analysis should be also provided.
Response: More detailed description of the statistical analysis had been added.
Results: Please provide some details about how relative expression of mRNA and proteins were obtained.
Response: Some details about how obtaining relative expression of mRNA and proteins had been added in the Materials and Methods section.
Please revise the paragraph describing the results about immunohistochemistry. In the current version of the manuscript this information is not adequate.
Response: Some details about immunohistochemistry had been added in the Materials and Methods section. Therefore, the paragraph describing the results about immunohistochemistry may be adequate.
Discussion: Although relevant information is included, this section is repetitive. The authors have made a very simplistic approach to this part of the work. Again, despite the importance of the topic and the interesting results obtained, the authors have made little effort to discuss these results in an adequate way to arouse the interest to the scientific community. As in the rest of the manuscript, it seems that the authors have not dedicated to this manuscript the necessary work and effort to make it scientifically relevant.
Response: Discussion section had been revised. Please see the Discussion section.
Conclusion: Page 7; Lines 250 and follows: I would recommend to revise also the conclusion section. In its present form, it is just a brief description of the obtained results and it does not make clear the relevance that the results obtained could have on a practical level on ovine reproduction.
Response: Conclusion section had been revised. Please see the Conclusion section.

Reviewer 2 Report
In this manuscript, the authors report the modulation in the expression of different proteins from the Nod-like receptor family in the lymph nodes of ewes during early pregnancy.
The manuscript is well written, with novelty and overall good merit. Here are some concerns/questions.
- There are some important lack of informations in the Material and Methods.
a) You state that the rams were housed with females for the pregnant groups. How was the pregnancy assessed to ensure the proper selection of timepoints?
b) How and why exactly were those timepoints selected? Could you describe why?
c) Although minor, it is usually suggested to verify the results of qPCR with two housekeeping genes? Was it done too, or only with GAPDH in your project?
d) line 100 and 101, 'described previously' is repeated. Please correct.
e) For immunohistochemistry, why are only NOD2 and NLRP7 staining performed? Why not the other markers? Please explain.
f) How were performed the experiments (In duplicate, triplicate, etc.)
-Figure 2 (western), Y axis is presented as 'relative expression values' whereas in figure 1 (qPCR) it is presented as 'relative values'. Is there a reason for this disparity? If not, could you address to be consistend throughout the figures?
-Figure 3, I have some major concerns for this figure. The pictures are of relatively poor quality and the staining is not very convincing. Could you address that?
-Line 174, 'expression level in the villi'; I assume you're talking about the placental villi? If so, could you please juste add placental?
-Addressing the discussions through all the different targeted proteins is interesting. However, in all your paragraph, I think that your conclusion of each protein are often too speculative. for example: line 200-201 'upregulation of NOD2 is beneficial for modulation of innate immune responses'. Your research projet is fairly descriptive with few, if no functional of mechanistic experiments (which is fine). However, you have no overall results proving of suggesting that this upregulation is beneficial. I would recommend being less speculative. For example, line 213-214 seems better ; 'CIITA during pregnancy MAY be related'.
Thank you!
Author Response
Response to Reviewer 2 Comments
In this manuscript, the authors report the modulation in the expression of different proteins from the Nod-like receptor family in the lymph nodes of ewes during early pregnancy.
The manuscript is well written, with novelty and overall good merit. Here are some concerns/questions.
Response: Thanks
There are some important lack of informations in the Material and Methods.
- a) You state that the rams were housed with females for the pregnant groups. How was the pregnancy assessed to ensure the proper selection of timepoints?
Response: The information had been deleted. Please see the words with yellow background in Animals and experimental design subsection.
- b) How and why exactly were those timepoints selected? Could you describe why?
Response: The reasons had been added. Please see the words with green background in Animals and experimental design.
We analysed the expression of complement components at these specific days of DN16, DP13, 16 and 25. The reasons are that the changed expression of the Nod-like receptor family is due to pregnancy, and the main factors are progesterone and IFNT during early pregnancy. The average of oestrous cycle is 17 days in sheep. There were significantly higher concentrations of progesterone on days 12-13 in plasma, and lower progesterone concentrations on 15-16 during the luteal phase of the ovine oestrous cycle (Mcnatty et al., 1973). IFNT (Protein X) and additional proteins were detected between days 14 and 21 in sheep (Godkin et al., 1982).
There is no DN25, because the average of oestrous cycle is 17 days in sheep. DN13 is almost similar to DP13 according to above reasons.
The above reasons had been present in previous reference that is cited in this manuscript.
Godkin JD, Bazer FW, Moffatt J, et al. Purification and properties of a major, low molecular weight protein released by the trophoblast of sheep blastocysts at day 13-21. Journal of Reproduction & Fertility, 1982, 65(1):141.
Mcnatty KP, Revefeim KJ, Young A. Peripheral plasma progesterone concentrations in sheep during the oestrous cycle. Journal of Endocrinology, 1973, 58(2):219-225.
- c) Although minor, it is usually suggested to verify the results of qPCR with two housekeeping genes? Was it done too, or only with GAPDH in your project?
Response: It has reported that GAPDH can be used as a housekeeping gene in many human tissues (Barber et al., 2005), so GAPDH is used as a housekeeping gene in this study. In addition, we only compared mRNA expression in one tissue (lymph node).
Barber RD, Harmer DW, Coleman RA, Clark BJ. GAPDH as a housekeeping gene: analysis of GAPDH mRNA expression in a panel of 72 human tissues. Physiol Genomics. 2005;21(3):389-95.
- d) line 100 and 101, 'described previously' is repeated. Please correct.
Response: 'described previously' had been deleted. Please see the words with yellow background in Western blot analysis subsection.
- e) For immunohistochemistry, why are only NOD2 and NLRP7 staining performed? Why not the other markers? Please explain.
Response: NOD2 was the representative for NODs and NLRP7 was the representative for NLRPs. In addition, other proteins had almost the same location pattern. Please see the words with yellow background in Immunohistochemistry analysis subsection.
- f) How were performed the experiments (In duplicate, triplicate, etc.)
Response: ‘n = 6 for each group, and each group consisted of three replicates’. Please see the words with green background in Animals and experimental design and Statistical analysis subsections.
Figure 2 (western), Y axis is presented as 'relative expression values' whereas in figure 1 (qPCR) it is presented as 'relative values'. Is there a reason for this disparity? If not, could you address to be consistend throughout the figures?
Response: It had been consistend throughout the figures with relative expression values. Please see the Figure 2 and Figure 2.
Figure 3, I have some major concerns for this figure. The pictures are of relatively poor quality and the staining is not very convincing. Could you address that?
Response: Some details about immunohistochemistry had been added in the Materials and Methods section. The results about immunohistochemistry were according to immunostaining intensities.
Line 174, 'expression level in the villi'; I assume you're talking about the placental villi? If so, could you please juste add placental?
Response: ‘placental’ had been added. Please see the words with green background in the Discussion section.
Addressing the discussions through all the different targeted proteins is interesting. However, in all your paragraph, I think that your conclusion of each protein are often too speculative. for example: line 200-201 'upregulation of NOD2 is beneficial for modulation of innate immune responses'. Your research projet is fairly descriptive with few, if no functional of mechanistic experiments (which is fine). However, you have no overall results proving of suggesting that this upregulation is beneficial. I would recommend being less speculative. For example, line 213-214 seems better ; 'CIITA during pregnancy MAY be related'.
Response: Discussion section had been revised. Please see the Discussion section.

Round 2
Reviewer 1 Report
Thank you to the authors for addressing the changes suggested. In general the manuscript have been modified very adequately, but however, the changes performed in the introduction section are insufficient. These changes are very superficial. The authors have only introduced two or three new sentences that do not respond to what was requested. Therefore, the introduction section continues to be a weak point of the manuscript.
In one of the added sentences (lines 42 and 43 of the new manuscript) it is necessary to include a bibliographical reference that supports this statement.
Despite the fact that the modifications in the introduction are not satisfactory, as it is mentioned above, the work has been revised and improved considerably and I consider that it can be accepted for publication after some minor revisions that consist of including the previously requested bibliographical reference and, if it is possible to better focus the introduction to give more information about NLRs and the importance of studying them in lymph nodes.
Author Response
Thank you to the authors for addressing the changes suggested. In general the manuscript have been modified very adequately, but however, the changes performed in the introduction section are insufficient. These changes are very superficial. The authors have only introduced two or three new sentences that do not respond to what was requested. Therefore, the introduction section continues to be a weak point of the manuscript.
Response: Thanks. The Introduction section had been revised according Reviewer’s suggestion. Please see the Introduction section.
In one of the added sentences (lines 42 and 43 of the new manuscript) it is necessary to include a bibliographical reference that supports this statement.
Response: A reference had been added. Please the Introduction section.
Despite the fact that the modifications in the introduction are not satisfactory, as it is mentioned above, the work has been revised and improved considerably and I consider that it can be accepted for publication after some minor revisions that consist of including the previously requested bibliographical reference and, if it is possible to better focus the introduction to give more information about NLRs and the importance of studying them in lymph nodes.
Response: The information about NLRs in lymph nodes had been added. Please see the Introduction section.
Reviewer 2 Report
Thank you for your revised manuscript.
You thoroughly addressed my comments and questions and, after a second review, consider that the manuscript has been greatly improved by the modification and adds, especially in the material and method section.
It was a pleasure re-reading it.
For the IHC pictures, although I agree that there is staining and I am ok with your results, I would still reiterate that I think you could take some better pictures. There is a blue stain even in the clear area (where it should be white). Maybe doing a White Balance before taking the picture could help? It may even greatly improve your contrast for the IHC staining.
However, It is only a suggestion, and will not add further comments on that for acceptance.
Thanks again!
Author Response
Thank you for your revised manuscript.
You thoroughly addressed my comments and questions and, after a second review, consider that the manuscript has been greatly improved by the modification and adds, especially in the material and method section.
It was a pleasure re-reading it.
Response: Thanks
For the IHC pictures, although I agree that there is staining and I am ok with your results, I would still reiterate that I think you could take some better pictures. There is a blue stain even in the clear area (where it should be white). Maybe doing a White Balance before taking the picture could help? It may even greatly improve your contrast for the IHC staining.
However, It is only a suggestion, and will not add further comments on that for acceptance.
Response: We had improved contrast for the IHC staining. Please see the Figure 3.